# Assessing the Reliability and Validity of a Questionnaire Evaluating Medical Students’ Attitudes, Knowledge, and Perceptions of Antibiotic Education and Antimicrobial Resistance in University Training

**DOI:** 10.3390/antibiotics13121126

**Published:** 2024-11-23

**Authors:** Olalla Vázquez-Cancela, Guillermo Lens-Perol, Marta Mascareñas-Garcia, Magdalena Santana-Armas, Juan Manuel Vazquez-Lago

**Affiliations:** 1Department of Preventive Medicine and Public Health Service, University Hospital of Santiago de Compostela, Rua da Choupana s/n, 15705 Santiago de Compostela, Spain; 2Health Research Institute of Santiago de Compostela (IDIS), 15706 Santiago de Compostela, Spain

**Keywords:** validity, reliability, attitudes, knowledge, perceptions, medical students, antimicrobial resistance

## Abstract

The misuse and overuse of antibiotics represent a critical global issue and one of the most pressing public health challenges of the 21st century. Training future healthcare professionals effectively is essential for ensuring responsible antibiotic use. This study aimed to validate a questionnaire designed to evaluate the knowledge, attitudes, and perceptions of medical students regarding the education they receive on infectious diseases, antimicrobial resistance, and antibiotic stewardship during their university studies. **Methods**: A self-administered questionnaire was developed and distributed to medical students at the University of Santiago de Compostela. Comprising 44 items, the questionnaire assessed eight key dimensions: “infection diagnosis”, “criteria for not prescribing antibiotics”, “initial antibiotic therapy”, “re-evaluation of therapy”, “quality of care”, “communication skills”, “antibiotic resistance”, and “teaching methodology”. Validation was carried out in two stages: Phase 1 involved content and face validity, while Phase 2 focused on reliability analysis. **Results**: A total of 295 students completed the questionnaire, with a mean age of 23.15 ± 1.78 years. The sample included 86 male (29.2%) and 209 female (70.8%) respondents. Content and face validity were established by a nominal group of five experts and a focus group of medicine and pharmacy students to ensure consensus on item understanding in the Spanish language. The questionnaire demonstrated high internal consistency with a Cronbach’s alpha of 0.92 and satisfactory item discrimination. Construct validity was confirmed through principal component analysis, which supported the presence of the eight predefined dimensions. **Conclusions**: The validated questionnaire exhibited strong reliability and validity, making it a valuable tool for assessing medical students’ training in antibiotic-related topics. Its application will enable the identification of areas for improvement in university curricula, ultimately contributing to the promotion of appropriate antibiotic use and the reduction of antimicrobial resistance.

## 1. Introduction

Antibiotics are essential drugs for treating bacterial infections and have saved countless lives since their discovery [1]. However, the rise of antimicrobial resistance (AMR) poses a severe public health threat, leading to increased morbidity, mortality, and significant economic costs [2]. AMR means that commonly used antibiotics are no longer effective against certain infections, as some bacteria have developed resistance to nearly all available antibiotics [3]. As these pathogens reproduce, they pass on these resistance traits, making infections more challenging to treat [4]. Recent data show that AMR has a devastating global impact on both developed and developing countries, causing around 5 million deaths associated with multidrug-resistant bacterial infections, including 1.27 million deaths directly attributable to bacterial AMR [5,6]. In Europe, these infections result in approximately 33,000 deaths annually, with an economic impact of 1.5 billion euros, while in Spain, an estimated 3000 deaths occur each year due to antibiotic-resistant infections [7]. Alarmingly, these figures are expected to increase significantly in the coming years [4,5,6].

The World Health Organization (WHO) has issued a warning that, if not addressed swiftly, antimicrobial resistance could lead to as many as 10 million deaths worldwide by 2050. This alarming projection underscores the necessity for comprehensive strategies like the One Health approach, which recognizes the interdependence between human, animal, and environmental health [8]. As early as 2006, the WHO set a goal to ensure that patients receive appropriate medications based on their clinical requirements, with the right dosages and durations, while maintaining affordable costs for both individuals and society [9]. In response to this challenge, the WHO launched the Global Action Plan on Antimicrobial Resistance in 2015. The plan emphasizes raising awareness, enhancing research, implementing infection prevention practices, promoting the responsible use of antibiotics, and optimizing resource management to encourage investment in new treatments [10].

One of the main drivers of increasing bacterial resistance is the improper use of antibiotics, such as incorrect prescriptions, self-medication, and incomplete treatment courses [1,11,12]. Addressing this problem is crucial to control the spread of resistance and safeguard the effectiveness of existing antibiotics for future generations [13]. This can be achieved by enhancing the training of medical students, ensuring that future prescribers are equipped to provide accurate guidance on antibiotic use. Given the key role of physicians in prescribing antibiotics, it is vital to assess the preparedness of medical students as future healthcare providers [14,15,16,17]. The European initiative Student-PREPARE examined medical students’ attitudes, knowledge, and perceptions regarding their education on antibiotics and antimicrobial resistance through a validated questionnaire [14]. Research shows that medical students globally often exhibit knowledge gaps concerning antibiotics and resistance [14,17,18,19]. Thus, evaluating their readiness is essential for ensuring that future physicians can contribute effectively to antimicrobial stewardship. In Spain, the medical program spans six years and includes relevant instruction on antibiotics through subjects such as microbiology, pharmacology, systemic infectious diseases and clinical microbiology, clinical pharmacology and pharmacotherapy, and preventive medicine and public health, which are integrated from the second to fifth years [20]. Physicians in the Spanish healthcare system play a crucial role in regulating antibiotic use in both community and hospital settings [21]. Factors like age and gender can influence the acquisition of knowledge and attitudes toward health topics, underscoring the importance of considering these variables in educational strategies [22,23]. Addressing gaps in medical students’ training could lead to improved prescribing practices and a reduction in antimicrobial resistance rates in Spain, which are currently above the European average [24,25]. Despite this need, there is a lack of comprehensive tools to assess medical students’ knowledge, attitudes, and perceptions regarding their education on antibiotics and resistance. Developing such tools would help to identify areas for improvement and guide enhancements in the academic training of future prescribers [26,27,28].

Thus, our objective was to validate the Spanish translation of a questionnaire that evaluates medical students’ knowledge, attitudes, and perceptions regarding their undergraduate education on antibiotics and antimicrobial resistance.

## 2. Results

### 2.1. Demographic Characteristics of Participants

A total of 295 fifth-year medical students, out of 349 invited, completed the questionnaire, yielding a response rate of 84.5%. All respondents who accepted the invitation participated fully in this study. The average age of participants was 23.15 ± 1.78 years. The sample included 86 male students (29.2%) and 209 female students (70.8%), all in their fifth year of medical school. Additionally, 22 respondents (7.7%) indicated that they were not citizens of the country where they were pursuing their medical studies.

### 2.2. Validation and Reliability of the Questionnaire

Step 1. Questionnaire Content and Face Validity

Content validity was evaluated by a nominal group of five experts from the Preventive Medicine and Public Health Service at the Clinic Hospital of Santiago de Compostela. Based on the input provided by this group, one item from the original version of the PREPARE questionnaire—“I feel qualified to identify situations in which it is indicated to use several antibiotics in combination”—that evaluated aspects related to antibiotic prescription indications was removed. The decision to eliminate this question was based on the group’s priority to ensure that students are capable of identifying situations where the use of a single antibiotic is indicated and that it is administered in line with local protocols. Additionally, in our context, the selection of antibiotic combinations is not determined by the physician alone, but rather by a multidisciplinary team known as the “Program Team for the Optimization of Antibiotic Use” in conjunction with the patient’s attending physician [20]. Aside from this modification, the nominal group chose to retain the eight predefined dimensions established at the outset.

Face validity was then assessed through a focus group of medical and pharmacy students, who confirmed the questionnaire’s structure as established by the nominal group.

Step 2. Reliability Analysis

#### 2.2.1. Internal Consistency and Item Discrimination Analysis

The initial Cronbach’s alpha was 0.923. Following the first reliability analysis, no items were removed, as none showed a homogeneity index significantly lower than 0.2 (see Table 1).

Table 2 presents the correlation coefficients for item discrimination, all of which are statistically significant (*p* < 0.05).

#### 2.2.2. Construct Validity

Principal component analysis (PCA) was conducted, yielding a KMO value of 0.897 and a significant Bartlett’s test of sphericity (*p* < 0.01). The analysis identified eight components that collectively accounted for 61.56% of the total variance, reflecting the eight dimensions that were defined beforehand.

In Table 3, the contribution of each component to the overall scale value is shown, along with the respective Cronbach’s alpha values for each dimension.

To evaluate the robustness of the questionnaire structure, we conducted an item response theory (IRT) analysis (see Table 4).

### 2.3. Exploratory Findings of Questionnaire Responses

#### 2.3.1. Students’ Perceptions of Their Preparedness in the Skills Required for Effective Infection Diagnosis and Treatment

Table 5 showcases a detailed breakdown of the mean and standard deviation for each item in the first three dimensions, capturing how students perceive the quality of care, evaluate communication skills, and understand issues related to antibiotic resistance. For a clearer interpretation of the results, student responses were thoughtfully categorized into three groups: disagree, neutral, and agree.

#### 2.3.2. Teaching Strategies and Antibiotic Education at the Faculty Level

Table 6 showcases the average scores and standard deviations for each item within dimensions 4 and 5, providing a snapshot of how pharmacy students perceive the faculty’s approach to teaching methodologies and antibiotic education. To offer a clearer picture of these perspectives, responses were grouped into three categories: disagree, neutral, and agree, allowing for a more nuanced interpretation of the findings.

For Item 44, which asked, “How do you think training on antibiotic treatment and prudent antibiotic use can be improved?” and was collected as an open-ended response, the majority of students highlighted several key areas for improvement. One of the most frequently mentioned suggestions was to place a greater emphasis on clinical practice through the use of clinical cases and simulations that mirror real-world scenarios of antibiotic prescription. Additionally, many students expressed the need for more structured and schematic classes, with less emphasis on lengthy theoretical explanations that are not directly applicable to clinical practice. Another common recommendation was to incorporate more practical and realistic cases into the curriculum, allowing students to apply their theoretical knowledge in clinical settings and gain decision-making experience. Lastly, several respondents suggested increasing the number of workshops and seminars focused on the appropriate use of antibiotics, which would provide opportunities for discussion and deeper understanding of best practices in clinical settings.

## 3. Discussion

Our translation and adaptation of the PREPARE project questionnaire specifically for fifth-year medical students yielded a scale with strong reliability, solid validity indices, and an excellent fit to the data, establishing a robust tool for evaluating educational outcomes in medical curricula.

The questionnaire developed in this study was based on the adaptation of the instrument used in the European Student-PREPARE project, which has been widely validated to assess the attitudes, knowledge, and perceptions of medical students in more than 20 countries [17]. The present study used a rigorous methodological approach to ensure the validity and reliability of the questionnaire, according to guidelines for developing instrumental studies; the scale meets psychometric properties, making it a reliable tool with high internal consistency and item discrimination [29]. Additionally, the questionnaire proves valid, as confirmed by the PCA.

The analysis of medical students’ responses indicates that they perceive their training in antibiotics to be inadequate, particularly regarding essential areas such as selecting the appropriate antibiotic therapy, determining the treatment duration, and collaborating within multidisciplinary teams for antibiotic management in hospital settings. This perceived lack of preparedness may be attributed to the fragmented structure of the medical curriculum in Spain, which superficially covers infectious diseases and their treatments without dedicated courses that provide in-depth knowledge on these subjects. [30]. On the other hand, we observe that the vast majority of surveyed students feel confident in identifying and diagnosing infectious diseases, but not in managing the aspects related to their treatment. Moreover, they do not feel adequately prepared for their future practice as medical residents. This finding is consistent with previous studies [14,31]. This gap in university training could potentially lead to inappropriate antibiotic prescribing, as well as erroneous knowledge and beliefs [2,32]. Since the education and training of future prescribers is one of the main strategies to improve antibiotic use, this study highlights that there is room for improvement. The observed results, which did not differ significantly from those of the study by Sanchez-Fabra [14], conducted four years earlier, may be attributed to the organization of medical education in Spain, where the curriculum is structured around the study of diseases by systems and organs, without specific courses dedicated to infectious diseases and their treatments.

From the perspective of medical education, these findings underscore the importance of reviewing and adapting curricula to include more practical experiences, such as clinical simulations and case studies, which allow students to apply their knowledge in real-world situations. Additionally, incorporating dedicated modules on the rational use of antibiotics and antimicrobial resistance would help students to develop stronger skills in these critical areas. From a public health perspective, the results indicate that, although many medical students feel competent in diagnosing infections, they often do not feel adequately prepared to manage antibiotic treatments effectively, consistent with previous findings [33,34]. This is a major concern, as insufficient competency in antibiotic prescribing could lead to inappropriate use, further worsening antimicrobial resistance [35,36]. The findings align with previous studies that highlight gaps in medical students’ training in areas like empirical antibiotic therapy and the management of bacterial resistance [14,17,18,19,33,34,35,36].

It is crucial to consider that the use of the validated questionnaire in diverse educational settings and geographic regions, especially in low- and middle-income countries, may require adaptation due to differences in regulations, clinical practices, and infectious disease prevalence. Prior studies have indicated that assessment tools must be customized to reflect local conditions [37,38]. In regions with limited health infrastructure and resources, the questionnaire’s validity could be compromised if it does not align with local realities. Variations in infectious disease management and educational standards suggest that the questionnaire should be adjusted to remain relevant [39,40]. For example, topics and teaching strategies must be contextualized to match current regulations and specific local needs. Previous research confirms that adapting tools to cultural and regional contexts is essential for maintaining their validity and effectiveness [41]. Thus, further research is needed to evaluate the questionnaire’s reliability in different settings and refine it as necessary to optimize its global application in medical education [42].

### 3.1. Questionnaire Development

Likert scales are widely recognized as effective instruments for data collection in diverse fields, with extensive research supporting their utility, validity, and reliability in quantifying subjective phenomena [43,44]. In this study, the face validity—referring to the degree to which the instrument appears effective in measuring the intended construct [45]—and content validity—indicating the extent to which the scale captures a comprehensive representation of the domain [46]—were rigorously evaluated through expert panel reviews, thorough literature analysis, and the application of PCA. The internal consistency of the questionnaire was confirmed by high Cronbach’s alpha values across all components, culminating in an overall alpha coefficient of 0.923. IRT analysis of the questionnaire using Guttman scaling and the Mokken discrimination index, along with the analysis previously conducted and presented in this study, provides a comprehensive perspective on the quality of the items in assessing and differentiating skill levels within a population of medical students. Given the high level of academic training and assumed expertise in this population, the analysis places particular emphasis on how well each item aligns with advanced knowledge and captures nuances in skill variation.

### 3.2. Strengths and Limitations

One of the limitations of this study is the sample size and selection process. However, 295 participants are considered adequate for research focused on the development and validation of a questionnaire, as it aligns with recommendations for preliminary surveys or scale development [47]. The literature suggests that, for initial item analysis, a sample size between 50 and 100 subjects is often sufficient, but to enhance reliability, a ratio of 5 to 10 subjects per item, with a minimum of 300 participants, is generally advised [48]. The sample size in this study enabled the confirmation of construct validity using factorial methods such as PCA. To further address this issue, we employed a penalized PCA approach (Ridge PCA), which is designed for smaller sample sizes and helps to prevent overfitting. Additionally, we implemented bootstrapping techniques to ensure the robustness of our findings, thereby mitigating concerns associated with sample size limitations (Appendix A). The outcomes of these analyses are consistent with our initial findings, supporting the robustness of our results. Another limitation pertains to the measurement scale used. Likert-type scales are prone to certain biases [49], particularly central tendency bias caused by social desirability, where respondents tend to avoid selecting extreme responses. To reduce this potential bias, we utilized Likert scales with more than five categories, which function similarly to visual analog scales and are more effective in capturing subtle variations in responses [50,51]. Specifically, we used a seven-point Likert scale in Block 2 to capture a nuanced range of attitudes and perceptions, while a four-point Likert scale was applied in Block 3 to prompt more definitive responses regarding behaviors [52].

Another relevant methodological aspect was the use of Varimax rotation in factor analysis, which allowed for the identification of item clustering within the predefined dimensions [53,54]. This approach supported the theoretical structure of the questionnaire and confirmed the suitability of the instrument to assess antibiotic-related education among medical students. However, it should be noted that the validation was conducted in a single university, limiting the generalizability of the results to other institutions with different curricula and educational contexts. Overall, the methodological approach used was solid and appropriate for developing a reliable and valid instrument to evaluate medical students’ training in the use of antibiotics. Nevertheless, future studies should consider conducting additional analyses, such as convergent and discriminant validity, as well as temporal stability testing through test–retest analysis, to further strengthen the robustness of the instrument in various educational settings.

One of the main strengths of this study is the methodological rigor applied in the development and validation of the questionnaire, which ensured its ability to accurately assess medical students’ competencies related to antibiotic use. Content validity was evaluated through a panel of experts in Preventive Medicine and Public Health, who made adjustments to the questionnaire to ensure that the items adequately reflected the expected competencies in the field of antibiotic use and antimicrobial resistance. Additionally, face validity was assessed through a focus group composed of medical and pharmacy students, allowing for the verification of item comprehension and clarity. Furthermore, the use of principal component analysis, along with the assessment of internal consistency through Cronbach’s alpha coefficient and IRT, provides a solid foundation for the instrument’s validity and reliability. The Guttman scaling results showed a range of item difficulties, with some items displaying considerable challenge levels. For a population of medical students, this variability is beneficial as it allows the questionnaire to capture a spectrum of knowledge, from foundational concepts to more complex, specialized knowledge. The presence of high-difficulty items is appropriate, given the academic background of the respondents, as it allows for differentiation, even within a cohort assumed to possess a high baseline of knowledge [52]. The Mokken discrimination analysis, which utilizes Loevinger’s H index, offers insight into the questionnaire’s ability to distinguish among students with varying degrees of mastery in the subject matter. The majority of items displayed a moderate H index, indicating adequate, though not exceptional, discriminative power. Given the high baseline knowledge among medical students, moderate discrimination may limit the tool’s effectiveness in differentiating between subtle variations in skill or depth of understanding [52]. The Rasch analysis reinforces the invariance of difficulty parameters, providing a scale on which items are comparable, regardless of the specific sample [55]. The consistency in difficulty values suggests that the questionnaire measures a unidimensional construct, specifically knowledge and skills related to the use of antibiotics and antimicrobial resistance. This aspect is important in validating educational instruments, as it ensures that the items are aligned in their aim to measure a single attribute [56,57]. This model identified low-difficulty items that were accessible to most students.

The combined analysis of IRT and Rasch confirms that the questionnaire is a robust tool for assessing students’ knowledge and attitudes regarding antibiotics and antimicrobial resistance [57,58]. The variability in difficulty and discrimination ensures that the questionnaire is useful for evaluating both basic knowledge and distinguishing advanced skill levels. This combined methodology provides a solid foundation for future implementations and adaptations of the questionnaire in different educational and research contexts, allowing for adjustments based on the specific needs of each study group.

## 4. Materials and Methods

This study was conducted between February and May 2023 in Santiago de Compostela, a city in the northwest region of Galicia, Spain. This research focused on fifth-year medical students enrolled in the Faculty of Medicine at the University of Santiago de Compostela. To participate, students had to meet two criteria: (1) be enrolled in the Preventive Medicine and Public Health course within the medical degree program at the university, and (2) have completed the practical training component of this course at the Santiago University Hospital Complex. We chose fifth-year medical students because, by this stage, they have completed all coursework related to antibiotic use, providing a comprehensive foundation in this area. Additionally, we selected students enrolled in preventive medicine as participants because the recruitment took place during clinical sessions in the hospital’s preventive medicine service.

Participants were recruited by disseminating information about the study, which included its objectives and significance. Alongside this information, students were provided with an anonymous questionnaire. To minimize the risk of duplications, the questionnaires were self-administered by students when they attended a session at the hospital’s preventive medicine service. This setting provided a controlled environment where each participant completed the questionnaire in person, under the supervision of the study staff. Therefore, the likelihood of duplication was minimized, as each student had only one opportunity to complete the questionnaire during their visit. Participation was voluntary, and no incentives were offered [57].

The design and adaptation of the questionnaire followed the methodology used in a previous study conducted with university pharmacy students. The instrument was based on a validated questionnaire from the Student-PREPARE project, which assesses the knowledge, attitudes, and perceptions of medical students regarding education on antibiotics and antimicrobial resistance in more than 20 European countries. The Student-PREPARE project is coordinated by the ESGAP (European Study Group for Antimicrobial Stewardship) under the ESCMID (European Society of Clinical Microbiology and Infectious Diseases) [17]. The conceptual framework for our study is based in this questionnaire, which we aimed to adapt to our context. This framework guided the categorization of the eight key dimensions into knowledge, attitudes, and perceptions, as it was designed to capture both the cognitive understanding and the affective responses of students regarding AMR, antibiotic treatment, and infections. Specifically, dimensions classified under ‘knowledge’ focus on objective, factual information, while those under ‘attitudes and perceptions’ capture subjective beliefs and perspectives. For this study, the original questionnaire was translated into Spanish and adapted to the context of Spanish university students (Appendix A).

The initial version of the questionnaire was refined through a comprehensive literature review [14,18,19,26,27,28,30,31,32]. The final version of the questionnaire was structured into 8 dimensions, consisting of 46 questions distributed across three blocks:

Block 1 gathered sociodemographic data (items 1–6), utilizing a combination of open-ended and multiple-choice questions.

Block 2 evaluated students’ knowledge, attitudes, and perceptions regarding their education on antibiotics and antimicrobial resistance. This section was organized into seven dimensions:

Students’ perception of their preparedness for infection diagnosis (items 7–12);

Indications for avoiding antibiotic prescriptions (items 13–15);

Understanding of empirical antibiotic therapy (items 16–21);

Re-evaluation of antibiotic therapy (items 22–24);

Perceived quality of care (items 25–26);

Perceived preparedness in communication skills (items 27–28);

Knowledge regarding antibiotic resistance (items 29–32).

All items in this block were scored on a 7-point Likert scale.

Block 3 addressed students’ opinions on teaching methodologies and evaluated their perceptions of the instructional methods used (items 33–43), scored on a 4-point Likert scale. Additionally, two supplementary questions were included: item 44, which assessed the appropriateness of the language used in the training, and item 45, an open-ended question inviting students to suggest improvements for educational programs on antibiotic treatment and prudent use.

The questionnaire’s structure closely followed the original instrument used in the Student-PREPARE project, ensuring comparability while incorporating specific adaptations for the local academic context.

### 4.1. Questionnaire Dissemination

The questionnaire was distributed to 5th-year medicine students in several practical work sessions carried out by the students at the hospital. Three distributions were made: the first on February 2023; the second on March 2023; and the third on May 2023.

### 4.2. Statistical Analysis

A comprehensive statistical analysis was conducted to characterize the study sample. Data were analyzed using SPSS version 20 (IBM Corp., Armonk, NY, USA). Additional analyses were performed with R (R Foundation for Statistical Computing, Vienna, Austria). The distribution of variables was first examined using the Kolmogorov–Smirnov test to assess normality. For quantitative data, results were expressed as the mean and standard deviation, while categorical and discrete variables were summarized using frequencies and percentages. Variables in Block 2, measured on a 7-point Likert scale ranging from 1 (completely disagree) to 7 (completely agree), were further grouped into three categories: “disagree”, “neutral”, and “agree” to facilitate interpretation. Similarly, variables pertaining to teaching methodology and education on antibiotics were transformed into a 4-point Likert scale, ranging from 0 (completely disagree) to 3 (completely agree). To validate the questionnaire, a two-step process was undertaken (Figure 1) [59,60,61]:

Step 1. Content and face validity of the questionnaire

Content validity was evaluated through two stages: (1) development stage and judgment stage by a nominal group of experts from the Preventive Medicine and Public Health Service at the Clinic Hospital of Santiago de Compostela, consisting of questionnaire development specialists, which included assessment of grammar, syntax, organization, appropriateness, and logical sequence of statements [62,63]; (2) face validity was assessed via a focus group of 7 students, 3 medical students and 4 pharmacy students, to ensure consensus on item understanding in the Spanish language.

Step 2. Reliability analysis

Reliability analysis was conducted in two stages under classical test theory (CTT) focus: (1) Evaluation of internal consistency through Cronbach’s alpha calculation [64,65], and analysis of item discrimination capability. The homogeneity index was based on the Pearson correlation coefficient between the item score and sum of scores from other items. Items with significative homogeneity <0.2 were eliminated from the questionnaire, as they did not measure the same construct as the overall questionnaire items [59,60,64]. And (2) evaluation of the resulting model through PCA with the varimax rotation method. To select the components, we based our approach on the Kaiser Criterion. Components with 2 or more items correlating >0.4 were identified as relevant [66]. Prior to factorial analysis, the Kaiser–Meyer–Olkin (KMO) measure of sampling adequacy was calculated, complemented by Bartlett’s test of sphericity for verification. Additionally, we conducted a penalized PCA using Ridge PCA and bootstrapping PCA with varimax with the Kaiser’ rotation method to verify the robustness of the results. Additionally, IRT was employed to analyze the survey structure and assess the reliability and validity of the instrument. This analysis was complemented with a Rasch analysis [55,56].

## 5. Conclusions

The analysis of reliability and validity suggests that the questionnaire designed in this study can be effectively applied to other groups of medical students to evaluate their attitudes, knowledge, and perceptions regarding antibiotic use and bacterial resistance. This instrument proves to be especially useful for identifying potential areas for enhancement within existing curricula and teaching approaches, enabling educators to recognize specific deficiencies and implement targeted strategies to improve the training of future prescribers. By addressing these gaps, the questionnaire helps to better prepare students to use antibiotics responsibly and communicate effectively about antimicrobial resistance in their professional roles.

## Figures and Tables

**Figure 1 antibiotics-13-01126-f001:**
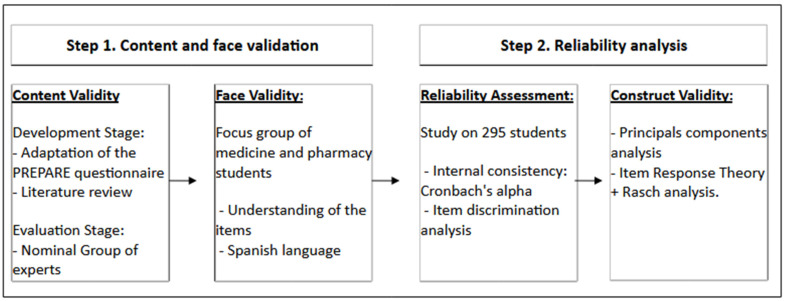
Flowchart of questionnaire development and validation.

**Table 1 antibiotics-13-01126-t001:** Scale item—total statistics.

Item (Variable)	Cronbach’s Alpha if Item Deleted	Homogeneity Index
Item 7: I feel able to recognize the clinical signs of infection	0.921	0.442
Item 8: I feel able to assess the clinical severity of infection (e.g., using criteria, such as the septic shock criteria)	0.921	0.463
Item 9: I feel able to use point-of-care tests (e.g., urine dipstick, rapid diagnostic tests for streptococcal pharyngitis)	0.922	0.390
Item 10: I feel able to interpret biochemical markers of inflammation (e.g., CRP)	0.921	0.459
Item 11: I feel able to decide when it is important to take microbiological samples before starting antibiotic therapy	0.921	0.487
Item 12: I feel able to interpret basic microbiological investigations (e.g., blood cultures, antibiotic susceptibility reporting)	0.920	0.547
Item 13: I feel able to identify clinical situations when not to prescribe an antibiotic	0.921	0.482
Item 14: I feel able to differentiate between bacterial colonization and infection (e.g., asymptomatic bacteriuria)	0.921	0.487
Item 15: I feel able to differentiate between bacterial and viral upper respiratory tract infections	0.920	0.545
Item 16: I feel able to select initial empirical therapy based on the most likely pathogen(s) and antibiotic resistance patterns, without using guidelines	0.919	0.616
Item 17: I feel able to decide the urgency of antibiotic administration in different situations (e.g., <1 h for severe sepsis, non-urgent for chronic bone infections)	0.919	0.614
Item 18: I feel able to prescribe antibiotic therapy according to national/local guidelines	0.920	0.575
Item 19: I feel able to assess antibiotic allergies (e.g., differentiating between anaphylaxis and hypersensitivity)	0.919	0.624
Item 20: I feel able to decide the shortest possible adequate duration of antibiotic therapy for a specific infection	0.919	0.634
Item 21: I feel able to prescribe using principles of surgical antibiotic prophylaxis	0.920	0.564
Item 22: I feel able to review the need to continue or change antibiotic therapy after 48–72 h, based on clinical evolution and laboratory results	0.919	0.693
Item 23: I feel able to assess clinical outcomes and possible reasons for failure of antibiotic treatment	0.918	0.685
Item 24: I feel able to decide when to switch from intravenous (IV) to oral antibiotic therapy	0.919	0.646
Item 25: I feel able to measure/audit antibiotic use in a clinical setting, and to interpret the results of such studies	0.920	0.567
Item 26: I feel able to work within the multidisciplinary team in managing antibiotic use in hospitals	0.919	0.623
Item 27: I feel to explain the importance of appropriate antibiotic use to patients and their families	0.919	0.628
Item 28: I feel to discuss antibiotic use and resistance issues effectively with healthcare professionals and team members	0.919	0.608
Item 29: I feel able to use knowledge of the common mechanisms of antibiotic resistance in pathogens	0.919	0.629
Item 30: I feel able to use knowledge of the epidemiology of bacterial resistance, including local/regional variations	0.919	0.597
Item 31: I feel able to practice effective Infection control and hygiene (to prevent spread of bacteria)	0.919	0.603
Item 32: I feel able to use knowledge of the negative consequences of antibiotic use (bacterial resistance, toxic/adverse effects, cost, Clostridium difficile infections)	0.919	0.601
Item 33: Faculty methodology: lectures with >15 people	0.924	0.120
Item 34: Faculty methodology: small group teaching with <15 people	0.923	0.234
Item 35: Faculty methodology: discussion of clinical cases and vignettes	0.922	0.397
Item 36: Faculty methodology: active learning assignments	0.922	0.371
Item 37: e-learning	0.923	0.284
Item 38: Faculty methodology: role-playing	0.923	0.253
Item 39: Faculty methodology: infectious diseases clinical placement	0.923	0.313
Item 40: Faculty methodology: microbiology clinical placement	0.923	0.224
Item 41: Faculty methodology: peer or near-peer teaching	0.923	0.237
Item 42: Overall, do you feel you have received sufficient teaching at medical school in antibiotic use for your future practice as a junior doctor?	0.926	−0.299
Item 43: Have any of your medical school examinations included questions on antibiotic treatment?	0.924	−0.141

**Table 2 antibiotics-13-01126-t002:** Correlation coefficients of the scale items.

Item (Variable)	Correlation Coefficient	*p*-Value
Item 7: I feel able to recognize the clinical signs of infection	0.477	<0.01
Item 8: I feel able to assess the clinical severity of infection (e.g., using criteria, such as the septic shock criteria)	0.500	<0.01
Item 9: I feel able to use point-of-care tests (e.g., urine dipstick, rapid diagnostic tests for streptococcal pharyngitis)	0.446	<0.01
Item 10: I feel able to interpret biochemical markers of inflammation (e.g., CRP)	0.498	<0.01
Item 11: I feel able to decide when it is important to take microbiological samples before starting antibiotic therapy	0.530	<0.01
Item 12: I feel able to interpret basic microbiological investigations (e.g., blood cultures, antibiotic susceptibility reporting)	0.587	<0.01
Item 13: I feel able to identify clinical situations when not to prescribe an antibiotic	0.522	<0.01
Item 14: I feel able to differentiate between bacterial colonization and infection (e.g., asymptomatic bacteriuria)	0.526	<0.01
Item 15: I feel able to differentiate between bacterial and viral upper respiratory tract infections	0.579	<0.01
Item 16: I feel able to select initial empirical therapy based on the most likely pathogen(s) and antibiotic resistance patterns, without using guidelines	0.646	<0.01
Item 17: I feel able to decide the urgency of antibiotic administration in different situations (e.g., <1 h for severe sepsis, non-urgent for chronic bone infections)	0.645	<0.01
Item 18: I feel able to prescribe antibiotic therapy according to national/local guidelines	0.613	<0.01
Item 19: I feel able to assess antibiotic allergies (e.g., differentiating between anaphylaxis and hypersensitivity)	0.656	<0.01
Item 20: I feel able to decide the shortest possible adequate duration of antibiotic therapy for a specific infection	0.664	<0.01
Item 21: I feel able to prescribe using principles of surgical antibiotic prophylaxis	0.599	<0.01
Item 22: I feel able to review the need to continue or change antibiotic therapy after 48–72 h, based on clinical evolution and laboratory results	0.718	<0.01
Item 23: I feel able to assess clinical outcomes and possible reasons for failure of antibiotic treatment	0.712	<0.01
Item 24: I feel able to decide when to switch from intravenous (IV) to oral antibiotic therapy	0.676	<0.01
Item 25: I feel able to measure/audit antibiotic use in a clinical setting, and to interpret the results of such studies	0.614	<0.01
Item 26: I feel able to work within the multidisciplinary team in managing antibiotic use in hospitals	0.664	<0.01
Item 27: I feel to explain the importance of appropriate antibiotic use to patients and their families	0.667	<0.01
Item 28: I feel to discuss antibiotic use and resistance issues effectively with healthcare professionals and team members	0.650	<0.01
Item 29: I feel able to use knowledge of the common mechanisms of antibiotic resistance in pathogens	0.659	<0.01
Item 30: I feel able to use knowledge of the epidemiology of bacterial resistance, including local/regional variations	0.630	<0.01
Item 31: I feel able to practice effective Infection control and hygiene (to prevent spread of bacteria)	0.638	<0.01
Item 32: I feel able to use knowledge of the negative consequences of antibiotic use (bacterial resistance, toxic/adverse effects, cost, Clostridium difficile infections)	0.637	<0.01
Item 33: Faculty methodology: lectures with >15 people	0.164	<0.01
Item 34: Faculty methodology: small group teaching with <15 people	0.283	<0.01
Item 35: Faculty methodology: discussion of clinical cases and vignettes	0.436	<0.01
Item 36: Faculty methodology: active learning assignments	0.414	<0.01
Item 37: e-learning	0.326	<0.01
Item 38: Faculty methodology: role-playing	0.296	<0.01
Item 39: Faculty methodology: infectious diseases clinical placement	0.365	<0.01
Item 40: Faculty methodology: microbiology clinical placement	0.261	<0.01
Item 41: Faculty methodology: peer or near-peer teaching	0.284	<0.01
Item 42: Overall, do you feel you have received sufficient teaching at medical school in antibiotic use for your future practice as a junior doctor?	−0.275	<0.01
Item 43: Have any of your medical school examinations included questions on antibiotic treatment?	−0.136	0.03

**Table 3 antibiotics-13-01126-t003:** Rotated component matrix and Cronbach’s alpha for the components.

Item (Variable)	Component							
	1	2	3	4	5	6	7	8
Item 20: I feel able to decide the shortest possible adequate duration of antibiotic therapy for a specific infection	0.757							
Item 21: I feel able to prescribe using principles of surgical antibiotic prophylaxis	0.718							
Item 17: I feel able to decide the urgency of antibiotic administration in different situations (e.g., <1 h for severe sepsis, non-urgent for chronic bone infections)	0.696							
Item 16: I feel able to select initial empirical therapy based on the most likely pathogen(s) and antibiotic resistance patterns, without using guidelines	0.675							
Item 24: I feel able to decide when to switch from intravenous (IV) to oral antibiotic therapy	0.585							
Item 18: I feel able to prescribe antibiotic therapy according to national/local guidelines	0.573							
Item 42: Overall, do you feel you have received sufficient teaching at medical school in antibiotic use for your future practice as a junior doctor?	−0.521							
Item 22: I feel able to review the need to continue or change antibiotic therapy after 48–72 h, based on clinical evolution and laboratory results	0.491							
Item 19: I feel able to assess antibiotic allergies (e.g., differentiating between anaphylaxis and hypersensitivity)	0.457							
Item 23: I feel able to assess clinical outcomes and possible reasons for failure of antibiotic treatment	0.434			0.430				
Item 8: I feel able to assess the clinical severity of infection (e.g., using criteria, such as the septic shock criteria)		0.750						
Item 7: I feel able to recognize the clinical signs of infection		0.733						
Item 10: I feel able to interpret biochemical markers of inflammation (e.g., CRP)		0.722						
Item 11: I feel able to decide when it is important to take microbiological samples before starting antibiotic therapy		0.666						
Item 12: I feel able to interpret basic microbiological investigations (e.g., blood cultures, antibiotic susceptibility reporting)		0.567						
Item 9: I feel able to use point-of-care tests (e.g., urine dipstick, rapid diagnostic tests for streptococcal pharyngitis)		0.537						
Item 28: I feel to discuss antibiotic use and resistance issues effectively with healthcare professionals and team members			0.852					
Item 26: I feel able to work within the multidisciplinary team in managing antibiotic use in hospitals			0.827					
Item 27: I feel to explain the importance of appropriate antibiotic use to patients and their families			0.794					
Item 25: I feel able to measure/audit antibiotic use in a clinical setting, and to interpret the results of such studies			0.632					
Item 31: I feel able to practice effective Infection control and hygiene (to prevent spread of bacteria)				0.803				
Item 32: I feel able to use knowledge of the negative consequences of antibiotic use (bacterial resistance, toxic/adverse effects, cost, Clostridium difficile infections)				0.739				
Item 29: I feel able to use knowledge of the common mechanisms of antibiotic resistance in pathogens	0.441			0.611				
Item 30: I feel able to use knowledge of the epidemiology of bacterial resistance, including local/regional variations				0.520				
Item 37: e-learning					0.661			
Item 40: Faculty methodology: microbiology clinical placement					0.652			
Item 39: Faculty methodology: infectious diseases clinical placement					0.645			
Item 38: Faculty methodology: role-playing					0.638			
Item 36: Faculty methodology: active learning assignments					0.614			
Item 41: Faculty methodology: peer or near-peer teaching					0.470			
Item 13: I feel able to identify clinical situations when not to prescribe an antibiotic						0.708		
Item 14: I feel able to differentiate between bacterial colonization and infection (e.g., asymptomatic bacteriuria)						0.658		
Item 15: I feel able to differentiate between bacterial and viral upper respiratory tract infections		0.472				0.528		
Item 33: Faculty methodology: lectures with >15 people							0.729	
Item 34: Faculty methodology: small group teaching with <15 people							0.701	
Item 43: Have any of your medical school examinations included questions on antibiotic treatment?								0.708
Item 35: Faculty methodology: discussion of clinical cases and vignettes								−0.410
Cronbach’s Alpha	0.814	0.784	0.852	0.828	0.723	0.848	0.844	0.726
Cronbach’s Alpha of the total scale	0.923							

Extraction method: PCA. Rotation method: varimax with Kaiser. Convergence was achieved in 9 iterations.

**Table 4 antibiotics-13-01126-t004:** Parameters of item response theory analysis and Rasch analysis.

Item (Variable)	IRT		Rasch Analysis
	Difficulty Index *	Discrimination Index **	Difficulty Index
Item 7: I feel able to recognize the clinical signs of infection	0.427	0.245	0.638
Item 8: I feel able to assess the clinical severity of infection (e.g., using criteria, such as the septic shock criteria)	0.315	0.216	0.735
Item 9: I feel able to use point-of-care tests (e.g., urine dipstick, rapid diagnostic tests for streptococcal pharyngitis)	0.359	0.230	0.799
Item 10: I feel able to interpret biochemical markers of inflammation (e.g., CRP)	0.288	0.205	0.288
Item 11: I feel able to decide when it is important to take microbiological samples before starting antibiotic therapy	0.454	0.248	0.853
Item 12: I feel able to interpret basic microbiological investigations (e.g., blood cultures, antibiotic susceptibility reporting)	0.386	0.237	0.691
Item 13: I feel able to identify clinical situations when not to prescribe an antibiotic	0.349	0.227	0.477
Item 14: I feel able to differentiate between bacterial colonization and infection (e.g., asymptomatic bacteriuria)	0.336	0.224	0.496
Item 15: I feel able to differentiate between bacterial and viral upper respiratory tract infections	0.431	0.246	0.431
Item 16: I feel able to select initial empirical therapy based on the most likely pathogen(s) and antibiotic resistance patterns, without using guidelines	0.498	0.250	0.584
Item 17: I feel able to decide the urgency of antibiotic administration in different situations (e.g., <1 h for severe sepsis, non-urgent for chronic bone infections)	0.499	0.250	0.594
Item 18: I feel able to prescribe antibiotic therapy according to national/local guidelines	0.453	0.248	0.565
Item 19: I feel able to assess antibiotic allergies (e.g., differentiating between anaphylaxis and hypersensitivity)	0.387	0.237	0.597
Item 20: I feel able to decide the shortest possible adequate duration of antibiotic therapy for a specific infection	0.476	0.249	0.636
Item 21: I feel able to prescribe using principles of surgical antibiotic prophylaxis	0.454	0.248	0.565
Item 22: I feel able to review the need to continue or change antibiotic therapy after 48–72 h, based on clinical evolution and laboratory results	0.512	0.250	0.693
Item 23: I feel able to assess clinical outcomes and possible reasons for failure of antibiotic treatment	0.491	0.250	0.691
Item 24: I feel able to decide when to switch from intravenous (IV) to oral antibiotic therapy	0.465	0.248	0.621
Item 25: I feel able to measure/audit antibiotic use in a clinical setting, and to interpret the results of such studies	0.401	0.240	0.569
Item 26: I feel able to work within the multidisciplinary team in managing antibiotic use in hospitals	0.440	0.246	0.604
Item 27: I feel to explain the importance of appropriate antibiotic use to patients and their families	0.489	0.250	0.616
Item 28: I feel to discuss antibiotic use and resistance issues effectively with healthcare professionals and team members	0.466	0.248	0.582
Item 29: I feel able to use knowledge of the common mechanisms of antibiotic resistance in pathogens	0.476	0.249	0.632
Item 30: I feel able to use knowledge of the epidemiology of bacterial resistance, including local/regional variations	0.412	0.242	0.596
Item 31: I feel able to practice effective Infection control and hygiene (to prevent spread of bacteria)	0.452	0.248	0.598
Item 32: I feel able to use knowledge of the negative consequences of antibiotic use (bacterial resistance, toxic/adverse effects, cost, Clostridium difficile infections)	0.437	0.246	0.589
Item 33: Faculty methodology: lectures with >15 people	0.324	0.220	0.115
Item 34: Faculty methodology: small group teaching with <15 people	0.189	0.154	0.223
Item 35: Faculty methodology: discussion of clinical cases and vignettes	0.369	0.233	0.369
Item 36: Faculty methodology: active learning assignments	0.353	0.228	0.353
Item 37: e-learning	0.403	0.241	0.403
Item 38: Faculty methodology: role-playing	0.273	0.198	0.232
Item 39: Faculty methodology: infectious diseases clinical placement	0.375	0.200	0.274
Item 40: Faculty methodology: microbiology clinical placement	0.237	0.181	0.205
Item 41: Faculty methodology: peer or near-peer teaching	0.396	0.239	0.216
Item 42: Overall, do you feel you have received sufficient teaching at medical school in antibiotic use for your future practice as a junior doctor?	0.044	0.042	−0.262
Item 43: Have any of your medical school examinations included questions on antibiotic treatment?	0.003	0.003	−0.134

* Difficulty index of Guttman scaling (Mean of item); ** Simplified discrimination index of Mokken (Loevinger’s H index).

**Table 5 antibiotics-13-01126-t005:** Medical students’ perception of their level of preparation for the different skills necessary for the proper diagnosis and treatment of infections.

Dimension	Item (Variable)	M	SD	Agreement (%)	Neutral (%)	Disagreement (%)
Students’ perception of their preparedness for infection diagnosis	Item 7: I feel able to recognize the clinical signs of infection	5.15	1.33	42.3	56.3	1.4
	Item 8: I feel able to assess the clinical severity of infection (e.g., using criteria, such as the septic shock criteria)	4.70	1.42	31.2	64.0	4.8
	Item 9: I feel able to use point-of-care tests (e.g., urine dipstick, rapid diagnostic tests for streptococcal pharyngitis)	3.96	1.99	26.6	56.6	16.8
	Item 10: I feel able to interpret biochemical markers of inflammation (e.g., CRP)	4.63	1.47	28.7	64.0	7.3
	Item 11: I feel able to decide when it is important to take microbiological samples before starting antibiotic therapy	4.36	1.71	29.0	56.9	14.1
	Item 12: I feel able to interpret basic microbiological investigations (e.g., blood cultures, antibiotic susceptibility reporting)	4.08	1.69	21.5	63.0	15.6
Indications for avoiding antibiotic prescriptions	Item 13: I feel able to identify clinical situations when not to prescribe an antibiotic	4.17	1.54	20.6	68.0	11.3
	Item 14: I feel able to differentiate between bacterial colonization and infection (e.g., asymptomatic bacteriuria)	4.12	1.50	19.6	71.1	9.3
	Item 15: I feel able to differentiate between bacterial and viral upper respiratory tract infections	4.47	1.43	25.2	67.6	7.2
Understanding of empirical antibiotic therapy	Item 16: I feel able to select initial empirical therapy based on the most likely pathogen(s) and antibiotic resistance patterns, without using guidelines	2.71	1.51	5.9	44.6	49.5
	Item 17: I feel able to decide the urgency of antibiotic administration in different situations (e.g., <1 h for severe sepsis, non-urgent for chronic bone infections)	2.75	1.46	3.5	51.2	45.3
	Item 18: I feel able to prescribe antibiotic therapy according to national/local guidelines	3.44	1.69	12.5	58.5	28.9
	Item 19: I feel able to assess antibiotic allergies (e.g., differentiating between anaphylaxis and hypersensitivity)	3.85	1.59	14.6	69.1	16.3
	Item 20: I feel able to decide the shortest possible adequate duration of antibiotic therapy for a specific infection	2.77	1.51	5.3	50.7	44.0
	Item 21: I feel able to prescribe using principles of surgical antibiotic prophylaxis	3.10	1.54	5.7	56.9	37.5
Re-evaluation of antibiotic therapy	Item 22: I feel able to review the need to continue or change antibiotic therapy after 48–72 h, based on clinical evolution and laboratory results	3.33	1.45	7.2	64.4	28.4
	Item 23: I feel able to assess clinical outcomes and possible reasons for failure of antibiotic treatment	3.49	1.48	8.6	66.1	25.3
	Item 24: I feel able to decide when to switch from intravenous (IV) to oral antibiotic therapy	2.98	1.53	5.3	58.9	35.8
Perceived quality of care	Item 25: I feel able to measure/audit antibiotic use in a clinical setting, and to interpret the results of such studies	3.09	2.03	7.3	56.5	36.2
	Item 26: I feel able to work within the multidisciplinary team in managing antibiotic use in hospitals	2.62	1.95	7.6	45.4	47.0
Perceived preparedness in communication skills	Item 27: I feel to explain the importance of appropriate antibiotic use to patients and their families	3.06	1.93	6.6	58.4	35.0
	Item 28: I feel to discuss antibiotic use and resistance issues effectively with healthcare professionals and team members	2.76	2.02	8.2	50.2	41.6
Knowledge regarding antibiotic resistance	Item 29: I feel able to use knowledge of the common mechanisms of antibiotic resistance in pathogens	3.38	1.46	8.3	64.7	27.0
	Item 30: I feel able to use knowledge of the epidemiology of bacterial resistance, including local/regional variations	2.94	1.51	7.3	52.4	40.2
	Item 31: I feel able to practice effective Infection control and hygiene (to prevent spread of bacteria)	4.37	1.62	28.3	57.0	14.7
	Item 32: I feel able to use knowledge of the negative consequences of antibiotic use (bacterial resistance, toxic/adverse effects, cost, Clostridium difficile infections)	4.68	1.75	37.7	50.9	11.4

M: mean. SD: standard deviation.

**Table 6 antibiotics-13-01126-t006:** Medical students’ perceptions regarding teaching methodology and antibiotic training at the faculty.

Dimension	Item (Variable)	M	SD	Agreement (%)	Neutral (%)	Disagreement (%)
Opinions on teaching methodologies	Item 33: Faculty methodology: lectures with >15 people	0.73	1.34	11.4	11.4	77.2
	Item 34: Faculty methodology: small group teaching with <15 people	1.43	1.52	31.7	17.1	51.2
	Item 35: Faculty methodology: discussion of clinical cases and vignettes	2.41	1.35	58.8	21.1	20.1
	Item 36: Faculty methodology: active learning assignments	1.37	1.47	27.1	19.9	52.9
	Item 37: e-learning	1.00	1.34	15.1	18.9	66.0
	Item 38: Faculty methodology: role-playing	0.74	1.38	16.2	6.6	77.2
	Item 39: Faculty methodology: infectious diseases clinical placement	1.53	1.71	37.6	7.9	54.5
	Item 40: Faculty methodology: microbiology clinical placement	0.61	1.18	12.5	8.3	79.2
	Item 41: Faculty methodology: peer or near-peer teaching	1.15	1.47	30.5	8.9	60.6
	Item 42: Overall, do you feel you have received sufficient teaching at medical school in antibiotic use for your future practice as a junior doctor?	2.42	0.78	13.9	4.4	81.6
	Item 43: Have any of your medical school examinations included questions on antibiotic treatment?	1.01	0.18	99.7	0.3	0.0

M: mean. SD: standard deviation.

## Data Availability

Data availability is under petition.

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
