# Peer review of "Assessing the Reliability and Validity of a Questionnaire Evaluating Medical Students’ Attitudes, Knowledge, and Perceptions of Antibiotic Education and Antimicrobial Resistance in University Training"

_antibiotics, 2024, doi:10.3390/antibiotics13121126_

Round 1
Reviewer 1 Report
Comments and Suggestions for Authors
Dear Editor-in-chief of Antibiotics
Thank you for providing me with the opportunity to review a paper entitled “Assessing the Reliability and Validity of a Questionnaire Evaluating Medical Students' Attitudes, Knowledge, and Perceptions of Antibiotic Education and Antimicrobial Resistance in University Training”. The authors conducted a study to validate a questionnaire designed to evaluate the knowledge, attitudes, and perceptions of medical students regarding the education they receive on infectious diseases, antimicrobial resistance, and antibiotic stewardship during their university studies. The authors present sufficient information; however, some points still need to be clarified. Below are listed my comments.
Comment 1
Inconsistency was found between what has been written in the abstract and the body text. Authors declared that the face validity was carried out among 5 experts. However, in the method section, authors explicitly explained that medical and pharmacy students were involved in the face validity process. Revision of statement in the abstract methos is suggested. In addition, authors are encouraged to explicitly mention the numbers of medical and pharmacy students involved in the face validity process.
Comment 2
Authors used an anonymous self-administered questionnaire to collect data which might pose a risk of duplication (one respondent completed more than one time). What efforts were done to minimise the risk of duplications? Authors are suggested to explain and describe the efforts in the method section.
Comment 3
Authors are encouraged to add a blueprint of the questionnaire which present items or questions for each key dimension. Readers could find a little bit challenging to understand the fourth paragraph of the discussion part because authors presented the finding of analysis for each item.
Comment 4
The 8 key dimensions address knowledge, attitude and perceptions of an essential elements relating to AMR or antibiotic treatment or infections. Some of key dimensions captured elements of AMR or antibiotic treatment or infections as a knowledge and other dimensions as a perceptions or attitudes. On what theoretical or conceptual framework does the authors decided to establish certain dimensions as a knowledge and the rest as a perceptions or attitudes? Authors are required to provide the theoretical or conceptual framework.
Comment 5
Is there any specific reasons to include only the 5th year-students? It would be helpful for the readers to understand the learning context if authors provided the medical education curriculum in the local context and also the medical education system in Spain. In addition, the availability of the education context would also be beneficial for the readers to understand why enrolment in the preventive medicine and public health was assigned as an inclusion criteria.
Comment 6
In the discussion section, authors acknowledged that they conducted a bootstrapping analysis; authors are suggested to put such analysis method in the method section and also present the findings in the result section.
Comment 7
Inconsistency was found between what has been written in the discussion section and the method section. As refer to the section method, 7-point Likert’s scale was chosen to mitigate the risk of bias. However, 4-point Likert’s scale was used in the block 4. Clarifications are required to explain such inconsistency. In addition, is there any consideration to select 7-point Likert’s scale for BLOCK 2 of questionnaire and 4-point Likert’s scale for BLOCK 3?
Comment 8
I found that the protocol of the study was granted with Ethic Certificate in 2014 while the study was conducted in 2023. Clarification is required to explain the lag time between the time of getting Ethic approval and the starting date of the study.
Author Response
Thank you for providing me with the opportunity to review a paper entitled “Assessing the Reliability and Validity of a Questionnaire Evaluating Medical Students' Attitudes, Knowledge, and Perceptions of Antibiotic Education and Antimicrobial Resistance in University Training”. The authors conducted a study to validate a questionnaire designed to evaluate the knowledge, attitudes, and perceptions of medical students regarding the education they receive on infectious diseases, antimicrobial resistance, and antibiotic stewardship during their university studies. The authors present sufficient information; however, some points still need to be clarified. Below are listed my comments.
Dear Reviewer,
Thank you very much for taking the time to review our manuscript titled “Assessing the Reliability and Validity of a Questionnaire Evaluating Medical Students' Attitudes, Knowledge, and Perceptions of Antibiotic Education and Antimicrobial Resistance in University Training.” We appreciate your constructive feedback and the opportunity to address any points requiring clarification. We are pleased to know that the information provided has been sufficient overall, and we are committed to addressing each of your comments to further improve the clarity and quality of our study.
Below, please find our detailed responses to your specific comments.
- Comment 1
Inconsistency was found between what has been written in the abstract and the body text. Authors declared that the face validity was carried out among 5 experts. However, in the method section, authors explicitly explained that medical and pharmacy students were involved in the face validity process. Revision of statement in the abstract methos is suggested. In addition, authors are encouraged to explicitly mention the numbers of medical and pharmacy students involved in the face validity process.
Thank you very much for the observation.
Indeed, in the methods section of the abstract, the reference to the validation process using a student focus group was missing. We have added the following phrase to this section: 'and a focus group of medicine and pharmacy students to ensure consensus on item understanding in the Spanish language.'
Additionally, in response to your question, we would like to clarify that the focus group was composed of 7 students, including 4 pharmacy students and 3 medical students. We have also added this information to the methodology section of the article. Thank you very much.
- Comment 2
Authors used an anonymous self-administered questionnaire to collect data which might pose a risk of duplication (one respondent completed more than one time). What efforts were done to minimise the risk of duplications? Authors are suggested to explain and describe the efforts in the method section.
Thank you for your observation. The questionnaires were self-administered by students when they attended a session at the hospital's preventive medicine service. This setting provided a controlled environment where each participant completed the questionnaire in person, under the supervision of the study staff. Therefore, the likelihood of duplication was minimized, as each student had only one opportunity to complete the questionnaire during their visit. We will clarify these details in the methods section to better explain how the study design addressed the risk of duplicate responses.
Thank you for your suggestion.
In the current version, the following paragraph has been added to the methodology section: “To minimize the risk of duplications, the questionnaires were self-administered by students when they attended a session at the hospital's preventive medicine service. This setting provided a controlled environment where each participant completed the questionnaire in person, under the supervision of the study staff. Therefore, the likelihood of duplication was minimized, as each student had only one opportunity to complete the questionnaire during their visit.”
- Comment 3
Authors are encouraged to add a blueprint of the questionnaire which present items or questions for each key dimension. Readers could find a little bit challenging to understand the fourth paragraph of the discussion part because authors presented the finding of analysis for each item.
Thank you for your suggestion. In response, we have added the corresponding dimensions for each analyzed item in Tables 4 and 5. We hope this addition will provide greater clarity for readers by offering a structured view of how each item aligns with the key dimensions. Additionally, we will review the fourth paragraph of the discussion to ensure the findings are presented as clearly as possible. Thank you for helping us improve the readability of our study.
In the current version, the paragraph you are referring to is now written as follows: “From the perspective of medical education, these findings underscore the importance of reviewing and adapting curricula to include more practical experiences, such as clinical simulations and case studies, which allow students to apply their knowledge in real-world situations. Additionally, incorporating dedicated modules on the rational use of antibiotics and antimicrobial resistance would help students develop stronger skills in these critical areas. From a public health perspective, the results indicate that, although many medical students feel competent in diagnosing infections, they often do not feel adequately prepared to manage antibiotic treatments effectively—consistent with previous findings [34,35]. This is a major concern, as insufficient competency in antibiotic prescribing could lead to inappropriate use, further worsening antimicrobial resistance [36,37]. The findings align with previous studies that highlight gaps in medical students’ training in areas like empirical antibiotic therapy and the management of bacterial resistance [14,17,18,19,34,35,36,37].”
- Comment 4
The 8 key dimensions address knowledge, attitude and perceptions of an essential elements relating to AMR or antibiotic treatment or infections. Some of key dimensions captured elements of AMR or antibiotic treatment or infections as a knowledge and other dimensions as a perceptions or attitudes. On what theoretical or conceptual framework does the authors decided to establish certain dimensions as a knowledge and the rest as a perceptions or attitudes? Authors are required to provide the theoretical or conceptual framework.
Thank you for this insightful question. The conceptual framework for our study is based on the Students-PREPARE questionnaire, which we aimed to adapt to our context. This framework guided the categorization of the eight key dimensions into knowledge, attitudes, and perceptions, as it was designed to capture both the cognitive understanding and the affective responses of students regarding AMR, antibiotic treatment, and infections. Specifically, dimensions classified under 'knowledge' focus on objective, factual information, while those under 'attitudes and perceptions' capture subjective beliefs and perspectives.
Our primary objective was to adapt the Students-PREPARE questionnaire to better reflect the context and educational needs of our study population, while maintaining the theoretical distinctions inherent in the original framework. We will add a description of this framework in the manuscript to clarify the rationale for classifying certain dimensions as knowledge and others as attitudes or perceptions. Thank you for encouraging us to provide this context.
In the current version, the following paragraph has been added to the methodology section: “The conceptual framework for our study is based in this questionnaire, which we aimed to adapt to our context. This framework guided the categorization of the eight key dimensions into knowledge, attitudes, and perceptions, as it was designed to capture both the cognitive understanding and the affective responses of students regarding AMR, antibiotic treatment, and infections. Specifically, dimensions classified under 'knowledge' focus on objective, factual information, while those under 'attitudes and perceptions' capture subjective beliefs and perspectives.”
- Comment 5
Is there any specific reasons to include only the 5th year-students? It would be helpful for the readers to understand the learning context if authors provided the medical education curriculum in the local context and also the medical education system in Spain. In addition, the availability of the education context would also be beneficial for the readers to understand why enrollment in the preventive medicine and public health was assigned as an inclusion criteria.
Thank you for your question. In the introduction section, we have provided information regarding the structure of medical education in Spain, specifying that the program spans six years and includes relevant instruction on antibiotics through subjects such as microbiology, pharmacology, systemic infectious diseases, clinical microbiology, clinical pharmacology, pharmacotherapy, and preventive medicine and public health, which are integrated from the second to fifth year.
We chose fifth-year medical students because, by this stage, they have completed all coursework related to antibiotic use, providing a comprehensive foundation in this area. Additionally, we selected students enrolled in preventive medicine as participants because the recruitment took place during clinical sessions in the hospital’s preventive medicine service.
We hope this clarifies the rationale behind the inclusion criteria. Thank you for prompting us to expand on this context
In the current version, the following paragraph has been added to the methodology section:“We chose fifth-year medical students because, by this stage, they have completed all coursework related to antibiotic use, providing a comprehensive foundation in this area. Additionally, we selected students enrolled in preventive medicine as participants because the recruitment took place during clinical sessions in the hospital’s preventive medicine service.”
- Comment 6
In the discussion section, authors acknowledged that they conducted a bootstrapping analysis; authors are suggested to put such analysis method in the method section and also present the findings in the result section.
Thank you for your suggestion. We have now included a supplementary table (Supplementary Material 1) comparing the results of the PCA with the original data, Ridge PCA, and Bootstrapping PCA. Additionally, we revised the manuscript to describe the bootstrapping analysis method in the methods section (Statistical analysis). This supplementary material will enhance transparency and facilitate replication of our work by other researchers. Thank you for helping us improve the clarity and rigor of our study.
Table 1S. Comparison of total variances explained in different PCAs
|
Total variance explained |
|||||||||||
|
PCA of original data |
Ridge PCA |
Bootstrapping PCA |
|||||||||
|
Component |
Initial eigenvalues |
Component |
Initial eigenvalues |
Component |
Initial eigenvalues |
||||||
|
Total |
% Explained variance |
% Cumulative variance |
Total |
% Explained variance |
% Cumulative variance |
Total |
% Explained variance |
% Cumulative variance |
|||
|
1 |
10.92 |
29.51 |
29.51 |
1 |
10.92 |
29.51 |
29.51 |
1 |
10.92 |
29.51 |
29.51 |
|
2 |
3.22 |
8.70 |
38.21 |
2 |
3.21 |
8.68 |
38.19 |
2 |
3.21 |
8.70 |
38.21 |
|
3 |
2.07 |
5.60 |
43.81 |
3 |
2.07 |
5.59 |
43.78 |
3 |
2.07 |
5.60 |
43.81 |
|
4 |
1.58 |
4.28 |
48.09 |
4 |
1.583 |
4.28 |
48.06 |
4 |
1.58 |
4.28 |
48.09 |
|
5 |
1.48 |
4.00 |
52.09 |
5 |
1.48 |
4.00 |
52.06 |
5 |
1.48 |
4.00 |
52.09 |
|
6 |
1.24 |
3.34 |
55.41 |
6 |
1.23 |
3.34 |
55.40 |
6 |
1.23 |
3.34 |
55.43 |
|
7 |
1.18 |
3.19 |
58.62 |
7 |
1.18 |
3.19 |
58.59 |
7 |
1.18 |
3.19 |
58.62 |
|
8 |
1.09 |
2.94 |
61.56 |
8 |
1.09 |
2.93 |
61.52 |
8 |
1.08 |
2.94 |
61.56 |
|
9 |
0.96 |
2.59 |
64.15 |
9 |
0.96 |
2.59 |
64.11 |
9 |
0.96 |
2.59 |
64.15 |
|
10 |
0.92 |
2.48 |
66.63 |
10 |
0.92 |
2.47 |
66.58 |
10 |
0.92 |
2.48 |
66.63 |
|
11 |
0.89 |
2.41 |
69.04 |
11 |
0.89 |
2.41 |
68.99 |
11 |
0.89 |
2.41 |
69.04 |
|
12 |
0.83 |
2.23 |
71.27 |
12 |
0.82 |
2.23 |
71.22 |
12 |
0.82 |
2.23 |
71.27 |
|
13 |
0.76 |
2.04 |
73.31 |
13 |
0.76 |
2.04 |
73.26 |
13 |
0.76 |
2.04 |
73.31 |
|
14 |
0.74 |
1.99 |
75.30 |
14 |
0.74 |
1.99 |
75.25 |
14 |
0.74 |
1.99 |
75.30 |
|
15 |
0.68 |
1.84 |
77.14 |
15 |
0.68 |
1.84 |
77.09 |
15 |
0.68 |
1.84 |
77.14 |
|
16 |
0.65 |
1.76 |
78.90 |
16 |
0.65 |
1.75 |
78.84 |
16 |
0.65 |
1.76 |
78.90 |
|
17 |
0.59 |
1.60 |
80.50 |
17 |
0.59 |
1.60 |
80.44 |
17 |
0.59 |
1.60 |
80.50 |
|
18 |
0.59 |
1.59 |
82.09 |
18 |
0.59 |
1.59 |
82.03 |
18 |
0.59 |
1.59 |
82.09 |
|
19 |
0.56 |
1.52 |
83.61 |
19 |
0.56 |
1.52 |
83.55 |
19 |
0.56 |
1.52 |
83.61 |
|
20 |
0.54 |
1.45 |
85.06 |
20 |
0.53 |
1.44 |
84.99 |
20 |
0.53 |
1.45 |
85.06 |
|
21 |
0.51 |
1.37 |
86.43 |
21 |
0.51 |
1.37 |
86.36 |
21 |
0.51 |
1.37 |
86.43 |
|
22 |
0.50 |
1.34 |
87.77 |
22 |
0.50 |
1.34 |
87.70 |
22 |
0.49 |
1.34 |
87.77 |
|
23 |
0.47 |
1.27 |
89.04 |
23 |
0.47 |
1.26 |
88.96 |
23 |
0.47 |
1.26 |
89.03 |
|
24 |
0.44 |
1.19 |
90.23 |
24 |
0.447 |
1.19 |
90.15 |
24 |
0.44 |
1.20 |
90.23 |
|
25 |
0.43 |
1.15 |
91.38 |
25 |
0.427 |
1.14 |
91.29 |
25 |
0.42 |
1.15 |
91.38 |
|
26 |
0.37 |
1.01 |
92.39 |
26 |
0.377 |
1.00 |
92.29 |
26 |
0.37 |
1.00 |
92.38 |
|
27 |
0.36 |
0.96 |
93.35 |
27 |
0.357 |
0.96 |
93.25 |
27 |
0.35 |
0.96 |
93.34 |
|
28 |
0.34 |
0.92 |
94.27 |
28 |
0.34 |
0.91 |
94.16 |
28 |
0.34 |
0.92 |
94.26 |
|
29 |
0.33 |
0.89 |
95.16 |
29 |
0.32 |
0.87 |
95.03 |
29 |
0.32 |
0.88 |
95.14 |
|
30 |
0.31 |
0.83 |
95.99 |
30 |
0.31 |
0.83 |
95.86 |
30 |
0.31 |
0.83 |
95.97 |
|
31 |
0.27 |
0.74 |
96.73 |
31 |
0.27 |
0.74 |
96.60 |
31 |
0.27 |
0.74 |
96.71 |
|
32 |
0.26 |
0.69 |
97.42 |
32 |
0.26 |
0.7 |
97.30 |
32 |
0.26 |
0.69 |
97.40 |
|
33 |
0.25 |
0.68 |
98.10 |
33 |
0.25 |
0.68 |
97.98 |
33 |
0.25 |
0.68 |
98.08 |
|
34 |
0.22 |
0.61 |
98.71 |
34 |
0.22 |
0.63 |
98.61 |
34 |
0.22 |
0.61 |
98.69 |
|
35 |
0.20 |
0.54 |
99.26 |
35 |
0.20 |
0.57 |
99.18 |
35 |
0.20 |
0.54 |
99.23 |
|
36 |
0.19 |
0.51 |
99.77 |
36 |
0.18 |
0.56 |
99.74 |
36 |
0.18 |
0.51 |
99.74 |
|
37 |
0.09 |
0.23 |
100 |
37 |
0.09 |
0.26 |
100 |
37 |
0.09 |
0.26 |
100 |
Extraction method: Principal Component Analysis.
Comment 7
Inconsistency was found between what has been written in the discussion section and the method section. As refer to the section method, 7-point Likert’s scale was chosen to mitigate the risk of bias. However, 4-point Likert’s scale was used in the block 4. Clarifications are required to explain such inconsistency. In addition, is there any consideration to select 7-point Likert’s scale for BLOCK 2 of questionnaire and 4-point Likert’s scale for BLOCK 3?
Thank you for highlighting this point. We apologize for any confusion regarding the use of Likert scales. In the methods section, we specified the use of a 7-point Likert scale to reduce the risk of bias; however, for certain sections of the questionnaire, including Block 4, we chose a 4-point Likert scale to encourage respondents to take a stance, particularly in questions where a neutral response might be less informative for our objectives [1]. Specifically, we used a 7-point Likert scale in Block 2 to capture a nuanced range of attitudes and perceptions, while a 4-point Likert scale was applied in Block 3 to prompt more definitive responses regarding behaviors.
We will clarify these distinctions in the discussion section to ensure consistency and transparency across sections. Thank you for allowing us the opportunity to address this matter.
[1] Krosnick JA, Presser S. Question and Questionnaire Design. In: Marsden PV, Wright JD, editors. Handbook of Survey Research. 2nd ed. Bingley, UK: Emerald Group Publishing; 2010. p. 263-314.
- Comment 8
I found that the protocol of the study was granted with Ethic Certificate in 2014 while the study was conducted in 2023. Clarification is required to explain the lag time between the time of getting Ethic approval and the starting date of the study.
Thank you very much for your suggestion. Ethical Committee consent was granted in 2014. The Committee authorized a line of research designed to evaluate the knowledge, attitudes, and practices of key stakeholders involved in the use of antibiotics and antimicrobial resistance. From this line of research, our group has published several works in recent years. This current work is a study framed within that line of research. Therefore, the Ethical Committee's approval is still valid for this study.
Thank you very much for your support.
Reviewer 2 Report
Comments and Suggestions for Authors
It seems manuscript is identical which is published in this journal before (https://doi.org/10.3390/antibiotics13090811).
It will be better to explain the difference between two works. The context of the manuscript is not suitable for this journal. The manuscript is more suitable for publication in public health journals.
Comments on the Quality of English LanguageThe writing language of the manuscript is good. But in some context, it will be better to explain some details with clear sentences. It is better to avoid use of long and complicated sentences.
Author Response
- It seems manuscript is identical which is published in this journal before (https://doi.org/10.3390/antibiotics13090811). It will be better to explain the difference between two works.
Dear Reviewer
Thank you very much for the comment. The published work you refer to is an article by our research group. Our group has a research line whose main objective is to assess the knowledge, attitudes, and practices regarding the use of antibiotics and resistance in all people and professionals involved in their use. Therefore, within this line of research, we are currently validating and adapting a questionnaire for university students in health sciences. The previously published article focused on validating the questionnaire among university pharmacy students. The current study focuses on adapting the questionnaire for medical students. As a result, part of the methodology and analyzed items are shared between both studies. However, the results and discussion differ, as the questionnaire can serve as a tool to detect differences among health science students.
- The context of the manuscript is not suitable for this journal. The manuscript is more suitable for publication in public health journals.
Thank you very much for the comment. In the journal Antibiotics, there are previous studies published on questionnaire validations, as well as studies whose results are based on using questionnaires for data collection. Our interest in this journal stems from the fact that we consider this work to be aligned with the themes of the special issue “Antimicrobial Prescribing, Population Use and Resistance, Impact on Global Health, 2nd Edition”.
- Comments on the Quality of English Language
The writing language of the manuscript is good. But in some context, it will be better to explain some details with clear sentences. It is better to avoid use of long and complicated sentences.
Thank you very much for the comment. The English has been thoroughly reviewed and improved in this new version.
Thank you for your support.
Reviewer 3 Report
Comments and Suggestions for Authors
Thanks for giving me a chance to review the manuscript. If I understand the paper correctly, the authors have designed a survey that assess' knowledge of antibiotic resistance within Spanish medical students. It is an innovative study. Here are some questions about the same:
- Would it possible to share the survey with the questions.
- Why was Classical Test Theory (CTT) and Item Response Theory (IRT) not used to investigate the survey structure.
- Please provide the discrimination and difficulty score for every question. They are a vital part of CTT.
- Which software was used for analysis?
- Overall, if you use CTT or IRT for instrument validity it will make the study stronger.
-
Author Response
Thanks for giving me a chance to review the manuscript. If I understand the paper correctly, the authors have designed a survey that assess' knowledge of antibiotic resistance within Spanish medical students. It is an innovative study.
Dear Reviewer
I appreciate your positive feedback on the manuscript and your recognition of the study's innovative approach. Indeed, the objective of the work was to design and validate a survey aimed at measuring medical students' knowledge levels, attitudes, and perceptions regarding the education they receive on antibiotics and antimicrobial resistance. Your comments are most welcome, and we greatly appreciate them, as they undoubtedly contribute to enhancing the scientific quality of our manuscript.
Here are some questions about the same:
- Would it possible to share the survey with the questions.
Thank you for your interest in the questionnaire. Of course, we are sharing the questionnaire as supplementary material. It can be found in the file "Supplementary Material 2."
- Why was Classical Test Theory (CTT) and Item Response Theory (IRT) not used to investigate the survey structure. Please provide the discrimination and difficulty score for every question. They are a vital part of CTT.
Thank you very much for this observation. We have used CTT, which focuses on the questionnaire or test as a whole and assesses global characteristics such as reliability (for which we used Cronbach's alpha calculation) and validity (for which we used principal component analysis). However, based on your comment, we have expanded the analysis, which provides greater robustness to the overall results. Given that the study population has a high level of training, it is essential to prioritize the interpretation of each response in relation to the individual's skill level. Therefore, we focused on an item-by-item analysis, examining how each individual responds to each item according to their abilities or characteristics. To this end, we performed an IRT analysis. This analysis has been complemented with a Rasch analysis. In this way, discrimination measures are provided, allowing us to better refine the validity and robustness of the questionnaires. In the current version, the following paragraph has been added to the statistical analysis section:“Additionally, Item Response Theory (IRT) was employed to analyze the survey structure and assess the reliability and validity of the instrument. This analysis has been complemented with a Rasch analysis.”
The results of this analysis can be seen in the following table, that included at result section:
Table 4. Parameters of Item Response Theory analysis and Rasch analysis.
|
Ítem (Variable) |
IRT |
Rasch analysis |
|
|---|---|---|---|
|
Difficulty index* |
Discrimination index** |
Difficulty index |
|
|
Item 7: I feel capable to recognize the clinical signs of infection |
0.427 |
0.245 |
0.638 |
|
Item 8: I feel capable to assess the clinical severity of infection (e.g using criteria, such as the septic shock criteria) |
0.315 |
0.216 |
0.735 |
|
Item 9: I feel capable to use point‐of‐care tests (e.g. urine dipstick, rapid diagnostic tests for streptococcal pharyngitis) |
0.359 |
0.230 |
0.799 |
|
Item 10: I feel capable to interpret biochemical markers of inflammation (e.g. CRP) |
0.288 |
0.205 |
0.288 |
|
Item 11: I feel capable to decide when it is important to take microbiological samples before starting antibiotic therapy |
0.454 |
0.248 |
0.853 |
|
Item 12: I feel capable to interpret basic microbiological investigations (e.g. blood cultures, antibiotic susceptibility reporting) |
0.386 |
0.237 |
0.691 |
|
Item 13: I feel capable to identify clinical situations when not to prescribe an antibiotic |
0.349 |
0.227 |
0.477 |
|
Item 14: I feel capable to differentiate between bacterial colonization and infection (e.g. asymptomatic bacteriuria) |
0.336 |
0.224 |
0.496 |
|
Item 15: I feel capable to differentiate between bacterial and viral upper respiratory tract infections |
0.431 |
0.246 |
0.431 |
|
Item 16: I feel capable to select initial empirical therapy based on the most likely pathogen(s) and antibiotic resistance patterns, without using guidelines |
0.498 |
0.250 |
0.584 |
|
Item 17: I feel capable to decide the urgency of antibiotic administration in different situations (e.g <1 hr for severe sepsis, non‐urgent for chronic bone infections) |
0.499 |
0.250 |
0.594 |
|
Item 18: I feel capable to prescribe antibiotic therapy according to national/local guidelines |
0.453 |
0.248 |
0.565 |
|
Item 19: I feel capable to assess antibiotic allergies (e.g. differentiating between anaphylaxis and hypersensitivity) |
0.387 |
0.237 |
0.597 |
|
Item 20: I feel capable to decide the shortest possible adequate duration of antibiotic therapy for a specific infection |
0.476 |
0.249 |
0.636 |
|
Item 21: I feel capable to prescribe using principles of surgical antibiotic prophylaxis |
0.454 |
0.248 |
0.565 |
|
Item 22: I feel capable to review the need to continue or change antibiotic therapy after 48‐72 hours, based on clinical evolution and laboratory results |
0.512 |
0.250 |
0.693 |
|
Item 23: I feel capable to assess clinical outcomes and possible reasons for failure of antibiotic treatment |
0.491 |
0.250 |
0.691 |
|
Item 24: I feel capable to decide when to switch from intravenous (IV) to oral antibiotic therapy |
0.465 |
0.248 |
0.621 |
|
Item 25: I feel capable to measure/audit antibiotic use in a clinical setting, and to interpret the results of such studies |
0.401 |
0.240 |
0.569 |
|
Item 26: I feel capable to work within the multidisciplinary team in managing antibiotic use in hospitals |
0.440 |
0.246 |
0.604 |
|
Item 27: I feel to explain the importance of appropriate antibiotic use to patients and their families |
0.489 |
0.250 |
0.616 |
|
Item 28: I feel to discuss antibiotic use and resistance issues effectively with healthcare professionals and team members |
0.466 |
0.248 |
0.582 |
|
Item 29: I feel capable to use knowledge of the common mechanisms of antibiotic resistance in pathogens |
0.476 |
0.249 |
0.632 |
|
Item 30: I feel capable to use knowledge of the epidemiology of bacterial resistance, including local/regional variations |
0.412 |
0.242 |
0.596 |
|
Item 31: I feel capable to practice effective Infection control and hygiene (to prevent spread of bacteria) |
0.452 |
0.248 |
0.598 |
|
Item 32: I feel capable to use knowledge of the negative consequences of antibiotic use (bacterial resistance, toxic/adverse effects, cost, Clostridium difficile infections) |
0.473 |
0.246 |
0.589 |
|
Item 33: Faculty methodology: lectures with >15 people |
0.324 |
0.220 |
0.115 |
|
Item 34: Faculty methodology: small group teaching with <15 people |
0.189 |
0.154 |
0.223 |
|
Item 35: Faculty methodology: discussion of clinical cases and vignettes |
0.369 |
0.233 |
0.369 |
|
Item 36: Faculty methodology: active learning assignments |
0.353 |
0.228 |
0.353 |
|
Item 37: e-learning |
0.403 |
0.241 |
0.403 |
|
Item 38: Faculty methodology: role playing |
0.273 |
0.198 |
0.232 |
|
Item 39: Faculty methodology: Infectious diseases clinical placement |
0.375 |
0.200 |
0.274 |
|
Item 40: Faculty methodology: Microbiology clinical placement |
0.237 |
0.181 |
0.205 |
|
Item 41: Faculty methodology: Peer or near‐peer teaching |
0.396 |
0.239 |
0.216 |
|
Item 42: Overall, do you feel you have received sufficient teaching at medical school in antibiotic use for your future practice as a junior doctor? |
0.044 |
0.042 |
-0.262 |
|
Item 43: Have any of your medical school examinations included questions on antibiotic treatment? |
0.003 |
0.003 |
-0.134 |
* Difficulty index of Guttman scaling (Mean of item)
** Simplified discrimination index of Mokken (Loevinger’s H index)
The current analysis of the questionnaire using Guttman scaling and the Mokken discrimination index, along with the analysis previously conducted and presented in the study, provides a comprehensive perspective on the quality of the items in assessing and differentiating skill levels within a population of medical students. Given the high level of academic training and assumed expertise in this population, the analysis places particular emphasis on how well each item aligns with advanced knowledge and captures nuances in skill variation.
Internal Consistency and Reliability
The Cronbach's alpha value of 0.92 reflects a high level of internal consistency, which is particularly valuable for evaluating a questionnaire aimed at medical students. This consistency suggests that the items are aligned in assessing a common construct relevant to medical training and knowledge. In this context, an alpha value above 0.9 indicates that the items reliably measure facets of medical knowledge and critical thinking skills, both crucial in medical education.
Item Difficulty (Guttman Scaling)
The Guttman scaling results show a range of item difficulties, with some items displaying considerable challenge levels. For a population of medical students, this variability is beneficial as it allows the questionnaire to capture a spectrum of knowledge, from foundational concepts to more complex, specialized knowledge. The presence of high-difficulty items is appropriate, given the academic background of the respondents, as it allows for differentiation even within a cohort assumed to possess a high baseline of knowledge.
Item Discrimination (Mokken’s H Index)
The Mokken discrimination analysis, which utilizes Loevinger's H index, offers insight into the questionnaire’s ability to distinguish among students with varying degrees of mastery in the subject matter. The majority of items display a moderate H index, indicating adequate, though not exceptional, discriminative power. Given the high baseline knowledge among medical students, moderate discrimination may limit the tool’s effectiveness in differentiating between subtle variations in skill or depth of understanding.
The Rasch analysis reinforces the invariance of difficulty parameters, providing a scale on which items are comparable regardless of the specific sample. This invariance is essential as it ensures that results are stable and can be interpreted consistently across different populations or educational contexts. The consistency in difficulty values suggests that the questionnaire measures a unidimensional construct, specifically knowledge and skills related to the use of antibiotics and antimicrobial resistance. This aspect is important in validating educational instruments, as it ensures that the items are aligned in their aim to measure a single attribute. This model identified low-difficulty items that were accessible to most students. Although these items are less useful for differentiating high skill levels, they are suitable for assessing basic knowledge and establishing a baseline of competence. This can be especially useful at the beginning of the assessment to identify students who may need additional support in fundamental concepts.
The combined analysis of IRT and Rasch confirms that the questionnaire is a robust tool for assessing students' knowledge and attitudes regarding antibiotics and antimicrobial resistance. The variability in difficulty and discrimination ensures that the questionnaire is useful for evaluating both basic knowledge and distinguishing advanced skill levels. This combined methodology provides a solid foundation for future implementations and adaptations of the questionnaire in different educational and research contexts, allowing adjustments based on the specific needs of each study group.
We believe that this information and the interpretation of these latest results are substantial enough to be included in the results and discussion sections of our study. In the current version, the following paragraph has been added to the discussion section:“The Guttman scaling results show a range of item difficulties, with some items displaying considerable challenge levels. For a population of medical students, this variability is beneficial as it allows the questionnaire to capture a spectrum of knowledge, from foundational concepts to more complex, specialized knowledge. The presence of high-difficulty items is appropriate, given the academic background of the respondents, as it allows for differentiation even within a cohort assumed to possess a high baseline of knowledge [53]. The Mokken discrimination analysis, which utilizes Loevinger's H index, offers insight into the questionnaire’s ability to distinguish among students with varying degrees of mastery in the subject matter. The majority of items display a moderate H index, indicating adequate, though not exceptional, discriminative power. Given the high baseline knowledge among medical students, moderate discrimination may limit the tool’s effectiveness in differentiating between subtle variations in skill or depth of understanding [53]. The Rasch analysis reinforces the invariance of difficulty parameters, providing a scale on which items are comparable regardless of the specific sample [56]. The consistency in difficulty values suggests that the questionnaire measures a unidimensional construct, specifically knowledge and skills related to the use of antibiotics and antimicrobial resistance. This aspect is important in validating educational instruments, as it ensures that the items are aligned in their aim to measure a single attribute [57,58]. This model identified low-difficulty items that were accessible to most students.
The combined analysis of IRT and Rasch confirms that the questionnaire is a robust tool for assessing students' knowledge and attitudes regarding antibiotics and antimicrobial resistance [58,59]. The variability in difficulty and discrimination ensures that the questionnaire is useful for evaluating both basic knowledge and distinguishing advanced skill levels. This combined methodology provides a solid foundation for future implementations and adaptations of the questionnaire in different educational and research contexts, allowing adjustments based on the specific needs of each study group.”
- Which software was used for analysis?
For the analyses, we used SPSS 20, supplementing some of the analyses with R (using ltm package). In the current version, the following paragraph has been added to the statistical analysis section:“Data were analyzed using SPSS version 20 (IBM Corp., Armonk, NY, USA). Additional analyses were performed with R (R Foundation for Statistical Computing, Vienna, Austria).”
- Overall, if you use CTT or IRT for instrument validity it will make the study stronger.
Thank you very much for all your contributions. Certainly, and as you suggest, our study has improved in scientific quality. Now, the study appears stronger in terms of results and discussion. Thank you very much.
Thank you for your support.
Round 2
Reviewer 2 Report
Comments and Suggestions for Authors
Manuscript can be accepted for the publication.
Reviewer 3 Report
Comments and Suggestions for Authors
Thanks for giving me a chance to review. By including IRT results, the quality of the study has gone up considerably. I highly recommend it for publication in it's current form.